# Exploitation of Baird Aromaticity and Clar's Rule for Tuning the Triplet Energies of Polycyclic Aromatic Hydrocarbons

Felix Plasser 

Department of Chemistry, Loughborough University, Loughborough LE11 3TU, UK; f.plasser@lboro.ac.uk

**Abstract:** Polycyclic aromatic hydrocarbons (PAH) are a prominent substance class with a variety of applications in molecular materials science. Their electronic properties crucially depend on the bond topology in ways that are often highly non-intuitive. Here, we study, using density functional theory, the triplet states of four biphenylene-derived PAHs finding dramatically different triplet excitation energies for closely related isomeric structures. These differences are rationalised using a qualitative description of Clar sextets and Baird quartets, quantified in terms of nucleus independent chemical shifts, and represented graphically through a recently developed method for visualising chemical shielding tensors (VIST). The results are further interpreted in terms of a 2D rigid rotor model of aromaticity and through an analysis of the natural transition orbitals involved in the triplet excited states showing good consistency between the different viewpoints. We believe that this work constitutes an important step in consolidating these varying viewpoints of electronically excited states.

**Keywords:** aromaticity; antiaromaticity; chemical shift; hydrocarbons, benzene; $\pi$-conjugated systems



## 1. Introduction

Polycyclic aromatic hydrocarbons (PAHs) are a fascinating class of molecules providing versatile organic semi-conductors [1] for a variety of use cases such as photovoltaics [2] including singlet fission [3,4], charge transport [5], and field-effect transistors [6]. Due to their high tunability, PAHs and their derivatives are also particularly suitable for more intricate applications such as logic gates [7], single molecule conductance [8,9], thermoelectrics [10], and multi-stage redox systems [11]. Ever more intricate PAH structures are being synthesised [12,13] and ever more detailed spectroscopic experiments performed [14–16].

The properties of PAHs can be widely tuned using the bond topology [17–19] as well as by doping with heteroatoms [9,20–23]. To do so effectively, it is particularly helpful to establish clear structure-property relationships bridging between the molecular structure and the observable properties. As one option, this can be achieved by considering the biradical character and unpaired electrons [22,24–27]. A particularly attractive alternative option is given via aromaticity noting that, following Hückel's [28] and Clar's [29] rules, qualitative predictions of molecular properties can already be made following simple rules based on the bond topolgy and the number of electrons [18,19,22]. Even more Baird's rule [30] of excited state aromaticity allows to effectively predict triplet energies [31–34].

However, it is not trivial to move from the qualitative description of aromaticity based on bond topology to quantifiable aromaticity descriptors and a number of methods have been put forward for this task, e.g., the harmonic oscillator model of aromaticity [35], the aromatic fluctuation index [36], and the isomerisation stabilisation energies [37]. In this context, methods based on magnetic properties such as induced current densities [38–40] and nucleus independent chemical shifts (NICS) [41,42] have become particularly popular as direct probes of the electronic structure, which are, at the same time, connected to experiment via nuclear magnetic resonance spectroscopy. However, a special challenge

in the application of these methods is that the underlying quantities (the current density susceptibility and the chemical shielding) are both represented by tensor fields making it difficult to view all of the relevant information together. To meet this challenge, we have recently developed the VIST (visualisation of chemical shielding tensors) method [43] providing graphical information about the full chemical shielding tensor, thus, providing detailed information on (anti)aromaticity along with its local variations and anisotropy. The method is exemplified in Figure 1: The principal axes of the shielding tensor at a given point in space are, first, determined via an eigenvalue decomposition and subsequently visualised via dumb-bells whose size depends on the associated eigenvalue.

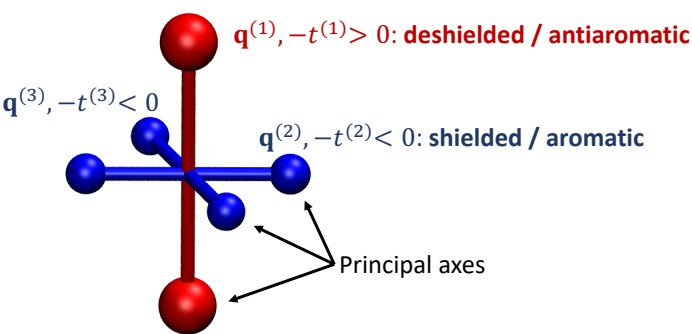

**Figure 1.** Visualisation of the chemical shielding tensor (VIST) via its three principal axes $\mathbf{q}^{(1)}$, $\mathbf{q}^{(2)}$, $\mathbf{q}^{(3)}$ and the associated eigenvalues $t^{(1)}$, $t^{(2)}$, $t^{(3)}$.

Using the VIST method, it is the purpose of this work to illustrate the concept of Baird aromaticity in a class of PAHs based on the biphenylene motif (see Figure 2). Recently, the biphenylene unit [44] has been considered for molecular conductance [8] and singlet fission [33]. Computational studies on the biphenylene unit integrated into larger PAHs have highlighted its effect on Baird aromaticity [45] and biradical character [46]. Here, we will investigate how minor structural variations in biphenylene derivatives can have a dramatic effect on triplet energies and analyse this effect in terms of bond topology, shielding tensors, and molecular orbitals. For this purpose, we construct three derivatives of biphenylene by formally fusing two benzene rings to the molecule. These molecules (**3**–**5**) along with biphenylene (**2**) and cyclobutadiene (**1**) are shown in Figure 2 (noting that the triplet energies and Baird aromaticity for molecules **2**–**5** has previously been investigated in [45]). Resonance structures of these molecules are shown with Clar sextets highlighted in blue and Baird quartets in red. Following the general rules laid out in [45], one can hypothesise that the structures with simultaneous Clar sextets and Baird quartets (i.e., **4** and **5**) will exhibit particularly pronounced Baird aromaticity leading to lowered triplet energies and we will elucidate these relations in detail based on energetic criteria, chemical shielding tensors, and molecular orbitals.

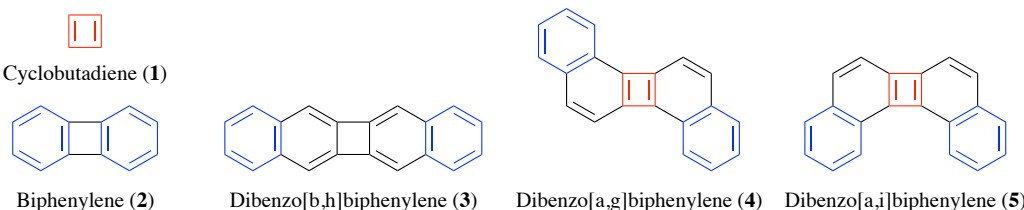

**Figure 2.** Chemical structures of the molecules considered within this work. Clar sextets are highlighted in blue, Baird quartets in red.

Within this work, we start the discussion with a theory section illustrating the concept of Baird aromaticity within the MO picture, discussing its effects on the chemical shielding, and providing the basics of the VIST method used in the graphical representation.

Subsequently, the computational results are presented starting with a basic consideration of energies, proceeding to a discussion of the shielding tensors in the CBD and benzene building blocks, and continuing with a detailed discussion of the properties of molecules **2–5** considering shielding tensors as well as molecular orbitals. Before concluding, we present a discussion of the relation between Baird aromaticity and other ways of describing excited-state electronic structure.

## 2. Theory

### 2.1. Aromaticity—The Rigid Rotor Viewpoint

In light of the discussion to follow, it is expedient to consider the phenomenon of aromaticity in the context of a simple 2D rigid rotor similarly to previously developed $\pi$-electron perimeter [47–49] and ring-current [50] models and following a related discussion based on Hückel theory [51]. The eigenstates of a 2D rigid rotor [52] are schematically represented in Figure 3b. The first state is non-degenerate whereas all other levels come in pairs of two degenerate states sorted according to an increasing value of the angular momentum quantum number $m_l$. The lowest state can be identified with the s-orbital ($m_l = 0$) of the hydrogen atom. The next levels have the same in-plane angular wavefunctions as the $p_x/p_y (|m_l| = 1)$, $d_{xy}/d_{x^2-y^2} (|m_l| = 2)$ pairs etc. increasing in energy and angular momentum.

Inspection of the molecular orbitals (MOs) of planar conjugated cyclic molecules shows that the MOs have similar symmetry properties as these rigid rotor eigenstates, i.e., there is one MO with no nodal plane, two MOs with one nodal plane, two MOs with two nodal planes etc. These MOs are shown in Figure 3a for benzene and (c,d) for cyclobutadiene (CBD) highlighting their similar structure among each other and with the idealised 2D rigid rotor. Only the highest, fully anti-bonding $\pi^*$-orbitals of benzene and CBD (not shown in Figure 3) do not fit the analogy, since they do not have a degenerate partner. Since the lowest level is non-degenerate and all others are doubly degenerate, it follows that one needs an odd number of electron pairs ($4n + 2$ electrons) to fill up any given level of degenerate orbitals, as shown in Figure 3a. On the other hand, if an even number of electron pairs ($4n$ electrons) are present, then the highest level will only be half-filled—usually accompanied by symmetry-breaking—as shown for CBD in Figure 3c. In line with general chemical knowledge, filled levels provide enhanced chemical stability and, hence, cyclic systems with $4n + 2$ electrons are particularly stable and systems with $4n$ electrons are not. This is the essence of Hückel's rule.

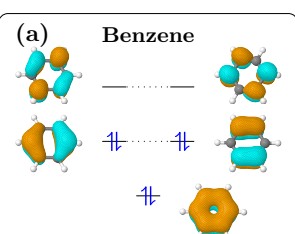
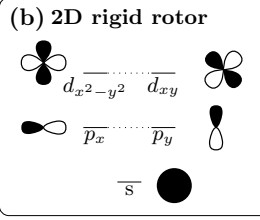
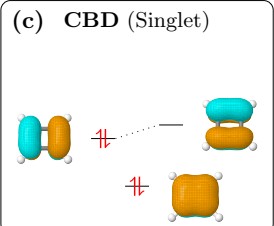
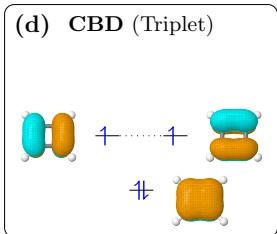

**Figure 3.** The concept of aromaticity illustrated using the 2D rigid rotor whose eigenstates are shown in (**b**). Benzene (**a**) and cyclobutadiene (**c,d**) are shown, highlighting the similarity between the $\pi$-MOs in these systems and the idealised rigid rotor eigenstates. With singlet multiplicity filled levels are obtained with $4n + 2$ electrons (**a**) whereas with $4n$ electrons (**c**) the HOMO and LUMO are quasidegenerate deriving from the same rigid rotor level. The rules are reversed for the triplet, which obtains filled shells with $4n$ electrons (**d**).

In Figure 3d, the case for the lowest triplet state in CBD is shown. An electron was promoted from the HOMO to the LUMO leading to two singly occupied orbitals, whose degeneracy is restored as CBD obtains four-fold symmetry. The occupation patterns now look similar to benzene in the sense that both quasidegenerate orbitals are evenly occupied. Indeed, CBD in its triplet state is considered aromatic according to Baird's rule [30], which states a reversal between aromaticity and antiaromaticity in the triplet state. Viewed

in the context of MO theory one expects a low triplet excitation energy due to the low HOMO/LUMO gap between the two quasidegenerate orbitals in CBD. However, due to the high degree of spatial overlap between HOMO and LUMO inducing enhanced exchange repulsion [53–55], one expects singlet excitation energies to be appreciably higher. The combination of low triplet energies with high exchange splitting provides a possibility of realising the energetics required for the singlet fission process [3] and, indeed, Baird aromaticity has been considered as a promising route toward new molecules capable of singlet fission [32–34,56].

### 2.2. Definition and Interpretation of the Chemical Shielding Tensors

The magnetic shielding, also known as chemical shielding, at a given point in space $\mathbf{R}$ is given as a tensor $\underline{\sigma}(\mathbf{R})$ defined by the relation

$$\mathbf{B}_{ind}(\mathbf{R}) = -\underline{\sigma}(\mathbf{R})\mathbf{B}_{ext} \tag{1}$$

where $\mathbf{B}_{ext}$ and $\mathbf{B}_{ind}$ are vectors describing the applied external and induced field, respectively, [57]. Thus, roughly speaking, the shielding is the negative of the ratio between external and induced field. In chemical systems, the induced field tends to be about a million times smaller than the external field and, hence, the shielding is on the order of a few ppm. More specifically, the shielding is given as a tensor field represented by a $3 \times 3$ matrix

$$\underline{\sigma}(\mathbf{R}) = \begin{pmatrix} \sigma_{xx}(\mathbf{R}) & \sigma_{xy}(\mathbf{R}) & \sigma_{xz}(\mathbf{R}) \\ \sigma_{yx}(\mathbf{R}) & \sigma_{yy}(\mathbf{R}) & \sigma_{yz}(\mathbf{R}) \\ \sigma_{zx}(\mathbf{R}) & \sigma_{zy}(\mathbf{R}) & \sigma_{zz}(\mathbf{R}) \end{pmatrix} \tag{2}$$

where all matrix elements depend on the position $\mathbf{R}$.

The nucleus independent chemical shift (NICS) [41], which serves as a prominent aromaticity criterion [42,58], is—in its basic form—defined as the negative of the spherically averaged magnetic shielding

$$\text{NICS}(\mathbf{R}) = -\frac{1}{3}\text{tr}[\underline{\sigma}(\mathbf{R})] = -\frac{1}{3}(\sigma_{xx}(\mathbf{R}) + \sigma_{yy}(\mathbf{R}) + \sigma_{zz}(\mathbf{R})). \tag{3}$$

In practical computations, the shielding is usually evaluated as the mixed second derivative of the energy with respect to an external magnetic field $B_\beta$ and the nuclear magnetic moment $\mu_\gamma$ [59–61]

$$\sigma_{\gamma\beta} = \frac{\partial^2 E}{\partial B_\beta \partial \mu_\gamma} \tag{4}$$

where $\beta$ and $\gamma$ are Cartesian coordinates (x, y, z). Note that in Equation (4) and all following equations atomic units are employed. Practically evaluating this expression leads to the following form [62,63]

$$\sigma_{\gamma\beta}(\mathbf{R}) = \underbrace{\frac{\alpha^2}{2}\left\langle \Psi_0 \left| \frac{\mathbf{r}\cdot(\mathbf{r}-\mathbf{R})\delta_{\beta\gamma} - r_\beta(\mathbf{r}-\mathbf{R})_\gamma}{|\mathbf{r}-\mathbf{R}|^3} \right| \Psi_0 \right\rangle}_{\sigma_{\gamma\beta}^{\text{dia}}(\mathbf{R})} \underbrace{-\alpha^2 \sum_{I\neq 0} \frac{\langle \Psi_0| \hat{L}'_\gamma |\mathbf{r}-\mathbf{R}|^{-3} |\Psi_I\rangle \langle \Psi_I| \hat{L}_\beta |\Psi_0\rangle}{E_I - E_0}}_{\sigma_{\gamma\beta}^{\text{para}}(\mathbf{R})} \tag{5}$$

where $\alpha$ is the fine-structure constant, $\Psi_0/\Psi_I$ are the ground/excited state wavefunctions, $E_0/E_I$ are their energies, $\hat{L}_\beta/\hat{L}'_\gamma$ are angular momentum operators with respect to the gauge origin and the point probed, and $\mathbf{r} = (x, y, z)$ is the distance vector from the gauge origin.

The total shielding is usually considered a sum of two terms $\sigma_{\gamma\beta}^{\text{dia}}$ (diamagnetic) and $\sigma_{\gamma\beta}^{\text{para}}$ (paramagnetic). This distinction, however, is not unique and the relative magnitude of these terms depends on the gauge origin chosen [61]. For practical calculations, it is expedient to include the gauge definition directly within the atomic orbitals [59,64]. However, for the following qualitative discussion, we will use a different convention

setting the gauge origin to the point where the shielding tensor is computed ($\mathbf{R} = \mathbf{0}$). Using this convention, the diamagnetic term—exemplified by one of its diagonal components $\sigma_{zz}^{\text{dia}}$—can be rewritten as

$$\sigma_{zz}^{\text{dia}}(\mathbf{0}) = \frac{\alpha^2}{2} \left\langle \Psi_0 \left| \frac{x^2 + y^2}{r^3} \right| \Psi_0 \right\rangle = \frac{\alpha^2}{2} \int \frac{\sin^2 \theta}{r} \rho(\mathbf{r}) d\mathbf{r} \geq 0 \qquad (6)$$

where $\rho(\mathbf{r})$ is the electron density, $r = |\mathbf{r}|$, and $\theta$ is the angle from the z-axis. This term, related to the Lamb formula, represents the textbook view of NMR spectroscopy—the chemical shielding is proportional to the electron density $\rho(\mathbf{r})$ in the vicinity of the nucleus of interest [52,63].

The second term $\sigma_{\gamma\beta}^{\text{para}}$, denoted paramagnetic, has a more complex form measuring how a perturbation of the wavefunction by an external magnetic field induces an angular momentum. It is usually given in a perturbative expansion in terms of the wavefunctions $\Psi_I$ and energies $E_I$ of the excited states of the system [60,62] as shown in Equation (5). Viewing, again, only one diagonal component at the gauge origin, we obtain

$$\sigma_{zz}^{\text{para}}(\mathbf{0}) = -\alpha^2 \sum_{I \neq 0} \frac{\langle \Psi_0 | \hat{L}_z/r^3 | \Psi_I \rangle \langle \Psi_I | \hat{L}_z | \Psi_0 \rangle}{E_I - E_0}. \qquad (7)$$

Whereas, the diamagnetic shielding given by Equation (6) is a non-specific term present whenever there is electron density, one finds that the paramagnetic shielding (Equation (7)) depends on the precise shapes and energies of the orbitals involved.

It is possible to construct a semi-quantitative model for the shielding contributions in (anti)aromatic systems based on the 2D rigid rotor discussed in Section 2.1. For this purpose, we assume that the electron density is distributed in a ring of radius $R_0$ around the gauge origin ($r = R_0$). In addition we assume this ring to be in the xy-plane, which means $\sin^2 \theta = 1$. Inserting these values into Equation (6), then leads to a diamagnetic shielding of

$$\sigma_{zz}^{\text{dia}} \approx \frac{\alpha^2}{2} \int \frac{1}{R_0} \rho(\mathbf{r}) d\mathbf{r} = \frac{\alpha^2 N_{el}}{2R_0} \geq 0 \qquad (8)$$

where $N_{el} = \int \rho(\mathbf{r}) d\mathbf{r}$ is the number of electrons contributing to the ring current.

In the case of an antiaromatic system, we can provide an estimate of the paramagnetic shielding by considering the HOMO/LUMO contribution. As shown in Figure 3c, the HOMO and LUMO of an antiaromatic system both relate to the same angular momentum quantum number $m_l$ (see also [51,58]). The corresponding real orbitals are given as $\pi^{-0.5} \cos(m_l \phi)$ and $\pi^{-0.5} \sin(m_l \phi)$ where $\phi$ is the azimuthal angle around the z-axis (cf. [52]). In addition, we can estimate the excitation energy of the first state via the HOMO/LUMO gap, i.e., $E_1$-$E_0 \approx E_{HL}$ (thus ignoring any post-MO contributions to the excitation energy [55]). Using these approximations, one obtains the following paramagnetic shielding.

$$\sigma_{zz}^{\text{para}} \approx -\alpha^2 \frac{\left| \left\langle \pi^{-0.5} \cos(m_l \phi) \left| -i \frac{\partial}{\partial \phi} \right| \pi^{-0.5} \sin(m_l \phi) \right\rangle \right|^2}{R_0{}^3 E_{HL}} = -\frac{\alpha^2 m_l{}^2}{R_0{}^3 E_{HL}} \leq 0 \qquad (9)$$

The analogous expression vanishes for aromatic systems as the matrix element of the angular momentum operator between functions with different $m_l$ vanishes under the assumptions stated. Note, that this is also true for a Baird aromatic system, as shown in Figure 3d, considering that due to the Pauli principle any excitation into a higher lying triplet state must involve orbitals of higher $m_l$.

The results are summarised in Table 1. For an aromatic system only the diamagnetic contribution plays a role and, hence, enhanced shielding going along with negative NICS values is expected at the centre of an aromatic ring. For an antiaromatic system diamagnetic shielding and paramagnetic deshielding both play a role and if the latter predominates,

then one finds the characteristic deshielding found at the centre of antiaromatic rings. Viewing Table 1, it is also interesting to discuss how the diamagnetic shielding changes with the ring size and number of electrons. If one assumes that with increasing ring size $R_0$, also the number of electrons $N_{el}$ increases, and specifically that $N_{el}/R_0$ is constant, then one finds that $\sigma_{zz}^{\text{dia}}$ should roughly stay constant. This consideration illustrates the suitability of NICS values to probe aromaticity in rings of various sizes [42,57]. Conversely, Table 1 also illustrates that, if $N_{el}$ is increased without increasing $R_0$, the diamagnetic shielding should go up. This explains why aromatic dianions are found to have significantly enhanced NICS values when compared to their aromatic dication counterparts [42,57]. Even more, if a large number of electrons are present in a small ring, then one expects large diamagnetic shielding even in the absence of aromaticity, as shown in the case of the $N_6H_6^{2+}$ ion [65].

**Table 1.** Diamagnetic and paramagnetic shielding contributions computed at the centre of a 2D rigid rotor in terms of the number of electrons ($N_{el}$), the radius ($R_0$), the angular momentum quantum number ($m_l$) of the HOMO and LUMO, and the HOMO/LUMO gap ($E_{HL}$).

| | Diamagnetic Shielding ($\sigma_{zz}^{\text{dia}}$) | Paramagnetic Deshielding ($\sigma_{zz}^{\text{para}}$) |
|---|---|---|
| Aromatic | $\dfrac{\alpha^2 N_{el}}{2R_0} \geq 0$ | $0$ |
| Antiaromatic | $\dfrac{\alpha^2 N_{el}}{2R_0} \geq 0$ | $-\dfrac{\alpha^2 m_l^2}{R_0^3 E_{HL}} \leq 0$ |

The above discussion suggests that NICS values can be viewed as a way to probe the symmetry properties of the frontier orbitals. Similar arguments also hold for multi-ring systems. If one can assume that each ring can be viewed as its own rigid rotor, then the NICS value reflects the local symmetry properties of this rigid rotor. However, the interpretation in multi-ring systems is more challenging due to the fact that the NICS value assigned to any given ring is not only affected by this ring but also by all other rings in the system. As a consequence, NICS values in multi-ring systems should be regarded with care, cf. [58], and not used as sole aromaticity criterion.

*2.3. Visualisation of Chemical Shielding Tensors (VIST)*

Equation (2) shows that the chemical shielding is a tensor field going over all space (cf. [60]), i.e., it is given by a $3 \times 3$ tensor containing 9 independent values at every point in space. Whereas scalar fields can be represented in 3D space via isosurfaces and vector fields via arrows, one has to construct a more involved representation for a tensor field. To do so, we have developed the VIST (visualisation of chemical shielding tensors) method [43]. VIST proceeds by computing the principal axes $\mathbf{q}^{(1)}$, $\mathbf{q}^{(2)}$, $\mathbf{q}^{(3)}$ of the shielding tensor via an eigenvalue decomposition

$$\underline{\sigma}\mathbf{q}^{(i)} = t^{(i)}\mathbf{q}^{(i)} \tag{10}$$

For visualisation, we construct a local coordinate system oriented according to the eigenvectors $\mathbf{q}^{(i)}$ and visualise the three components as dumb-bells whose size and length depends on the absolute value of the associated eigenvalue $|t^{(i)}|$ and whose color depends on the sign (blue or red), see Figure 1. Through encoding the eigenvectors and eigenvalues in the graphical representation, we are able to represent the full information given in the $3 \times 3$ tensor graphically. Specifically, we draw the length $L$ of the axis and the radius $R$ of the sphere as

$$L = 2 \times 0.3\sqrt{|t^{(i)}|} \qquad\qquad R = 0.03\sqrt{|t^{(i)}|} \tag{11}$$

where $t^{(i)}$ is given in ppm and $L$ and $R$ are given in Å. To compare these results to the NICS values, it is worth noting that in analogy to Equation (3) the NICS value is a third of the sum of the three eigenvalues according to

$$\text{NICS} = -\frac{1}{3}\text{tr}[\underline{\sigma}] = -\frac{1}{3}\left(t^{(1)} + t^{(2)} + t^{(3)}\right) \tag{12}$$

Reviewing Equation (1), we find that the principal axes of the shielding tensor are the directions in space where the induced magnetic field is parallel to the external magnetic field. If we set $\mathbf{B}_{ext} = B_0\mathbf{q}^{(i)}$ and insert this into Equation (1), we find by applying Equation (10) that

$$\mathbf{B}_{ind} = -\underline{\sigma}B_0\mathbf{q}^{(i)} = -t^{(i)}B_0\mathbf{q}^{(i)} = -t^{(i)}\mathbf{B}_{ext} \tag{13}$$

*2.4. Computational Details*

All molecular geometries were optimised using density functional theory (DFT) at the TPSSh/def2-TZVP level of theory [66–68]. Restricted Kohn–Sham theory (RKS) was employed for the singlet ground states ($S_0$) and unrestricted Kohn–Sham (UKS) for the first triplet states ($T_1$). Full spatial symmetry was employed, i.e., $D_{2h}$ for **1**–**3**, $C_{2h}$ for **4**, and $C_{2v}$ for **5**. In addition, time-dependent DFT (TDDFT) computations on the $T_1$ states were performed to obtain the natural transition orbitals (NTOs) [69,70]. These employed the range-separated CAM-B3LYP functional [71] to avoid known problems [72,73] in the TDDFT description of large conjugated $\pi$-systems along with the def2-TZVPP basis set. NTOs shown within this article were computed at the $S_0$ geometries; NTOs at the $T_1$ geometries are shown in Figures S1–S4 (Supplementary Materials). These computations were all carried out in Q-Chem [74–76].

Additional single-point energy computations on the TPSSh-optimised geometries were performed at the complete active space perturbation theory (CASPT2) level of theory [77] as implemented in OpenMolcas [78]. For the CASPT2 computations and the preceding complete active space self-consistent field (CASSCF) computations an active space of 12 electrons distributed in 12 $\pi$-orbitals was used. Following [79] no IPEA shift [80] was employed. The ANO-S-VDZP basis set was used [81]. NTOs between the $S_0$ and $T_1$ states (shown in Figures S1–S4) were computed following [82] using transition densities obtained via the state-interaction method [83]. Note that for single-state CASPT2, as employed here, the resulting NTOs are directly obtained from the CASSCF wavefunctions and there is no CASPT2-related correction.

Chemical shielding tensors for $S_0$ and $T_1$ states were computed using RKS and UKS, respectively, at the respective optimised geometries using TPSSh/def2-TZVP as implemented in Gaussian 09 [84] using gauge-including atomic orbitals [59] and applying tight SCF convergence criteria. The shielding tensors were analysed using the VIST (visualisation of chemical shielding tensors) method [43] employing a pre-release version of TheoDORE 2.4 [85]. The visual molecular dynamics (VMD) [86] program was used as a graphical backend for creating the tensor representations in connection with the molecular structures and NTO pictures.

The underlying research data are available via a separate repository [87]: Molecular coordinates; input/output files for Q-Chem, OpenMolcas, Gaussian, and TheoDORE.

## 3. Results

*3.1. Energetics*

We start with a discussion of the basic energetics of the molecules presented in Figure 2 considering their geometries optimised in their lowest singlet ($S_0$) and triplet ($T_1$) states at the DFT/TPSSh level. Table 2 shows the TPSSh energy values (with CASPT2 reference values given in parentheses). Starting with the adiabatic triplet excitation energy ($\Delta E(S_0/T_1)$), we find that it almost vanishes for cyclobutadiene (CBD, **1**) in line with CBD's expected antiaromaticity. Moving to biphenylene a substantial rise in triplet energy to

1.92 eV is seen. Interestingly, increasing the $\pi$-system to **3** further increases the triplet energy to 2.64 eV. Moving to the isomeric molecules **4** and **5** yields a dramatic drop in the excitation energy to just above 1 eV.

**Table 2.** Adiabatic excitation energies to the first triplet state $\Delta E(S_0/T_1)$ and total energies $E(S_0)$ and $E(T_1)$, referenced against the $S_0$ energy of **3**, for the molecules studied computed at the DFT/TPSSh level of theory (CASPT values in parentheses). All energies are given in eV.

| | Molecule | $\Delta E$ ($S_0/T_1$) | $E$ ($S_0$) | $E$ ($T_1$) |
|---|---|---|---|---|
| **1** | Cyclobutadiene | 0.133 (0.183) | - | - |
| **2** | Biphenylene | 1.920 (1.765) | - | - |
| **3** | Dibenzo[b,h]biphenylene | 2.638 (2.243) | 0.000 (0.000) | 2.638 (2.243) |
| **4** | Dibenzo[a,g]biphenylene | 1.032 (0.894) | 0.508 (0.594) | 1.541 (1.488) |
| **5** | Dibenzo[a,i]biphenylene | 1.008 (0.814) | 0.514 (0.598) | 1.522 (1.412) |

Due to the fact that molecules **3**, **4**, and **5** are isomeric, we can analyse their energetics in some more detail. In Table 2, the total energies of their singlet $E(S_0)$ and triplet $E(T_1)$ states, all referenced against the $S_0$ energy of **3**, are listed. This shows that the formal interconversion from **3** to **4** or **5** is strongly endothermic ($\approx$0.5 eV) in the singlet state. However, in the triplet state the energetic ordering is reversed meaning that **4** and **5** are stabilised by more than 1 eV with respect to **3**.

Owing to the antiaromatic character of the molecules involved and the ensuing low HOMO/LUMO gap it is not a priori clear whether the single reference description afforded by DFT is accurate, cf. [88–90]. Therefore, we have performed single-point computations at the multireference CASPT2 level to provide reference data on the energies involved. The CASPT2 energies, shown in parentheses in Table 2, do not only reproduce all the trends obtained with DFT but even show good quantitative agreement with most values within 0.15 eV between the two methods. The only larger deviation (0.4 eV) is obtained for the adiabatic excitation energy of **3**, an error, which we assign to the multiconfigurational character of its $T_1$ state, as discussed below. In any case, the agreement suggests that the chosen level of DFT/TPSSh provides a reliable description of the molecules studied.

The above-mentioned trends in energies clearly run counter to the standard picture stating that excitation energies should decrease with an extension of the $\pi$-system as confinement effects are reduced. Conversely, the concept of ground state antiaromaticity in connection with triplet state Baird aromaticity provides an attractive option for explanation. Reviewing the energetics in Table 2, we hypothesise that when moving from **1** to **3** the antiaromaticity and Baird aromaticity of CBD are both "blurred" as the $\pi$-system is extended leading to increased triplet energies. Conversely, reviewing Figure 2, we find that out of all structures considered only **4** and **5** possess resonance structures with a Baird quartet along with two Clar sextets. Following [45] we hypothesise that **4** and **5**, therefore, possess enhanced ground-state antiaromaticity as well as triplet state Baird aromaticity when compared to **3** explaining both the $S_0$ and $T_1$ energies in Table 2.

Simply looking at the energetics, it is difficult to assess whether (anti)aromaticity is really the underlying cause for the dramatic differences in energy or whether this is just a coincidence. A different viewpoint and more detailed insight is required to further substantiate the hypothesis. We will endeavour to provide this viewpoint via the computation of nucleus independent chemical shifts [42] and specifically our newly developed method for the visualisation of the underlying chemical shielding tensors [43].

### 3.2. Shielding Tensors for the Building Blocks

We start the discussion of shielding tensors with the building blocks, benzene and cyclobutadiene. In Figure 4, shielding tensors, computed at the TPSSh/def2-TZVP level, are shown 1 Å above the the centre of the ring. The tensor for benzene Figure 4a has a particularly simple shape with just one dominant out-of-plane component ($-29.5$ ppm)

and vanishing in-plane components (<1 ppm). This shape reflects the aromaticity of benzene. The dominant component agrees well with NICS(1)$_{zz}$ literature values computed at the B3LYP (-28.5 ppm) [57], PBE0 ($-29.5$ ppm) [43], MP2 ($-30.4$ ppm) [88], and CASSCF ($-28.8$ ppm) [88] levels highlighting that NICS values are robust descriptors of the electronic structure.

The shape of the shielding tensor is in qualitative agreement with the model of Equation (8) showing that a circular charge distribution should yield a strong out-of-plane contribution of the shielding tensor and vanishing in-plane components. Even more, we can approximate the numerical value. If we consider an effective radius of $R_0 = 2.65$ a.u. (1.4 Å) and assume that half the $\pi$-electrons contribute to the shielding effectively at the point probed ($N_{el} = 3$), then we can use Equation (8) to estimate the shielding as

$$- \sigma_{zz}^{\text{dia}} = -\frac{\alpha^2 \times 3}{2 \times 2.65} = -3.02 \times 10^{-5} = -30.2 \text{ ppm}. \tag{14}$$

This value is indeed in good agreement with the value of $-29.5$ ppm given in Figure 4a showing that the qualitative model developed is applicable to understand the physics involved. Note, however, that the extremely good match of less than 1 ppm deviation is just a coincidence.

Proceeding to CBD in its singlet state (Figure 4b), we find a strongly deshielded (red, +54.1 ppm) contribution representing the strong antiaromaticity in this system. CASSCF computations [88] place the corresponding NICS(1)$_{zz}$ value at a similar and only slightly lower value of +39.3 ppm highlighting that the method chosen performs reasonably well despite the expected multireference character. In parallel to the molecular plane, one slightly shielded (blue) and one slightly deshielded (red) contribution is found. This anisotropy of the shielding reflects the broken symmetry of CBD. The shielded component points toward the CC double bonds (bond length of 1.33 Å) and the deshielded component toward the single bonds (1.57 Å) illuminating the different properties of the two bonds.

Continuing with the qualitative model of Table 2, we again consider half the electrons as being involved ($N_{el} = 2$). Furthermore we set $m_l = 1$, $R_0 = 1.89$ a.u., and $E_{HL} = 0.076$ a.u. and we find by application of Equations (6) and (7) the following value for the shielding

$$- \sigma_{zz} = -\sigma_{zz}^{\text{dia}} - \sigma_{zz}^{\text{para}} = -\frac{\alpha^2 \times 2}{2 \times 1.89} + \frac{\alpha^2 \times 1^2}{1.89^3 \times 0.076} = -28.2 \text{ ppm} + 103.8 \text{ ppm} = +75.6 \text{ ppm} \tag{15}$$

This is, again, in qualitative agreement with the value of +54.1 ppm obtained in the computation highlighting that the viewpoint adopted here is a valid approximation of the rather involved shielding expressions.

Proceeding to the triplet state of CBD (Figure 4c), we find that the dominant component of the shielding becomes negative ($-17.6$ ppm) indicating Baird aromaticity for the triplet state. This phenomenon can be understood by considering Figure 3d: By promoting an electron from the HOMO to the quasidegenerate LUMO, the system now occupies a filled shell meaning enhanced stability. Reviewing Equation (7), we find that all orbitals up to $m_l = 1$ are filled and, therefore, no triplet excited states are present that could be coupled to the $T_1$ state via $\hat{L}_z$ meaning that the paramagnetic shielding at the ring centre vanishes. Thus, only the diamagnetic shielding $\sigma_{zz}^{\text{dia}}$ is left predicting a value of $-28.2$ ppm according to Equation (15), which again at least produces the same order of magnitude.

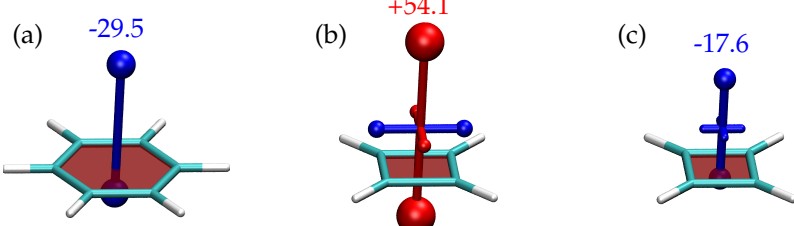

**Figure 4.** Visualisation of chemical shielding tensors (VIST) for (**a**) the $S_0$ state of benzene as well as (**b**) the $S_0$ and (**c**) the $T_1$ states of cyclobutadiene, all computed 1 Å above the molecular plane. The numbers (given in ppm) correspond to the negatives of the eigenvalues $t^{(i)}$ associated to the dominant out-of-plane contributions.

### 3.3. Shielding Tensors for Biphenylene Derivatives

Having discussed the individual benzene and CBD molecules, it is of interest to observe what happens when these are fused together obtaining the biphenylene molecule (**2**). Starting with the VIST plots for the singlet ground state (Figure 5a), we find that the antiaromaticity of the central CBD ring and the aromaticity of the outer benzene rings are both reduced in magnitude when compared to the building blocks. The reduction in the primary shielding components is modest for CBD (from +54.1 to +38.2 ppm) whereas a reduction by about two thirds occurs for the benzene components ($-29.5$ to $-11.1$ ppm).

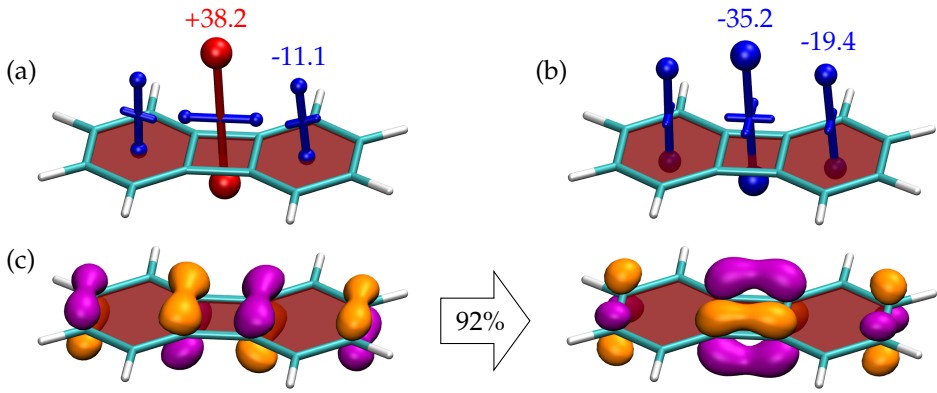

**Figure 5.** Visualisation of chemical shielding tensors (VIST) for (**a**) the $S_0$ and (**b**) the $T_1$ states of **2** computed 1 Å above the molecular plane. The dominant pair of natural transition orbitals (isovalue 0.07 a.u., weight 92 %) for the $S_0 \rightarrow T_1$ transition is shown in (**c**).

Next, we proceed to the $T_1$ state of biphenylene. TDDFT indicates that this state is dominated by the HOMO/LUMO transition with respect to the ground state. To obtain a rigorous graphical representation of the transition, we use the natural transition orbitals (NTOs) of the $S_0 \rightarrow T_1$ transition, shown in Figure 5c. The NTOs clearly reflect the shape of the HOMO and LUMO of CBD (see Figure 3). This indicates that the transition occurs largely between the two quasi-degenerate orbitals on CBD, which fits with the model of Baird aromaticity. The NTOs shown in Figure 5c, were computed at the $S_0$ geometry using the TDDFT/CAM-B3LYP level of theory. For comparison, we also show the TDDFT NTOs at the $T_1$ geometry as well as the CASPT2 NTOs at the $S_0$ and $T_1$ geometries (Figure S1). The shapes of all these NTOs are very similar highlighting that the assignment of the state character is robust with respect to, both, the geometry and the level of theory.

The $T_1$ shielding tensors, shown in Figure 5b, reflect at first glance the profound changes the system undergoes upon excitation and in agreement with [45] we find that the system becomes shielded/aromatic at all positions probed. The central CBD ring becomes strongly shielded with a value of $-35.2$ ppm that is higher than, both, isolated CBD in the triplet and benzene in the singlet state. Furthermore, the shielding at the benzene rings is

enhanced compared to the singlet. These changes highlight the Baird aromaticity of the triplet state in line with the NTOs presented.

Having established the properties of biphenylene, it is of interest to see how the local aromaticity changes upon addition of two benzene rings on the sides obtaining **3** as shown in Figure 6. Viewing the singlet ground state properties (Figure 6a), we find that the central biphenylene unit bears close resemblance to isolated biphenylene (Figure 5a) with the exception that the antiaromaticity at the central CBD unit is slightly reduced (to +27.2 ppm). The outer two benzene rings resemble isolated benzene showing strong shielding ($-24.4$ ppm).

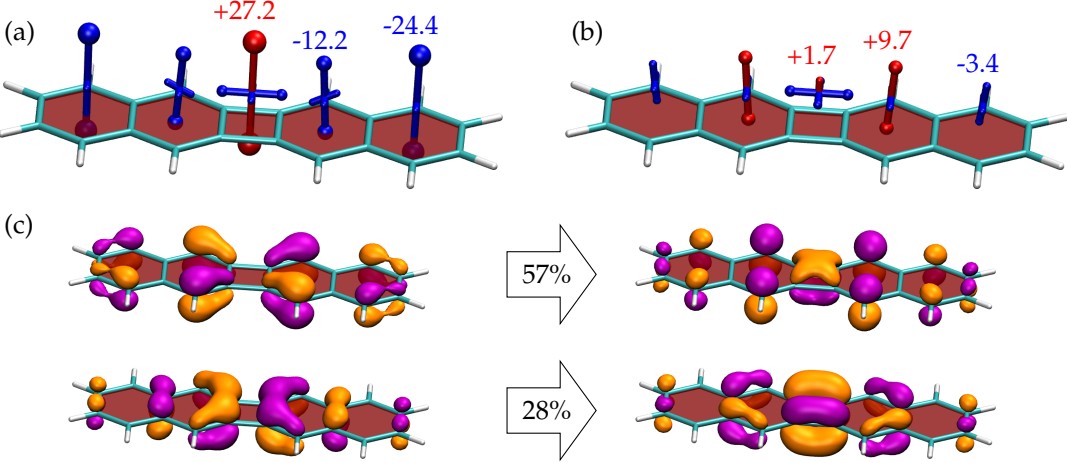

**Figure 6.** Visualisation of chemical shielding tensors (VIST) for (**a**) the $S_0$ and (**b**) the $T_1$ states of **3** computed 1 Å above the molecular plane. The two dominant pairs of natural transition orbitals (isovalue 0.05 a.u.) for the $S_0 \to T_1$ transition are shown in (**c**).

The shielding tensors for the lowest triplet state of **3** are shown in Figure 6b. These have a strikingly different appearance to the analogous values for biphenylene (Figure 5b). The central CBD and outer benzene rings have almost vanishing out-of-plane values (below 5 ppm in magnitude) whereas the intermediate benzene ring has a notably deshielded value of +9.7 ppm. To understand this behaviour, we turn to the associated NTOs as shown in Figure 6c. These NTOs differ in two important ways when compared to biphenylene (Figure 5c). First, we find that two NTO transitions contribute significantly to this state, whereas for biphenylene there was only one dominant transition. Second, we find that the dominant NTO pair, making up 57% of the overall transition has an entirely different shape when compared biphenylene. There is only little involvement of CBD and, interestingly, its orbitals contribute in the opposite way as expected. The acceptor orbital possesses some contribution of the fully bonding CBD $\pi$-orbital, shown at the bottom in Figure 3c. By contrast, the donor possesses some contribution of the fully anti-bonding CBD $\pi^*$-orbital (not shown in Figure 3). The dominant NTO pair, thus, highlights the strong mixing of the CBD MOs with the other MOs in the system. This consideration shows that this transition does not agree with the Baird aromaticity model laid out in Figure 3d. On the contrary, the NTOs on the two naphthalene units bear resemblance to the HOMO and LUMO of an isolated naphthalene molecule (see Figure S5) only that the density is somewhat concentrated toward the centre of the molecule. The emerging antiaromaticity can, therefore, be understood in the same sense that an individual napthalene molecule in its triplet state becomes antiaromatic [91]. Going back to Figure 6c, one finds that only the second NTO pair (contributing 28%) is of the expected shape analogous to biphenylene. The NTOs at the $T_1$ geometry (Figure S2) are similar to the ones shown in Figure 6c with the exception that the weights of the two NTO pairs are slightly altered. Similar shapes and weights for the NTOs are also found for CASPT2 (Figure S2).

In summary, we find that the difference in shielding tensors between **2** and **3** is well-reflected by the difference in the underlying NTOs highlighting the consistency between aromaticity descriptors and the MO picture. From a more methodological point of view we want to emphasise that the involvement of two NTO pairs in the state implies multi-configurational character of the triplet state [92,93] meaning that the single determinant description provided by UKS is not able to cover all relevant aspects of the wavefunction. Indeed CASSCF indicates strong multireference character for the triplet state with a weight of only 50% for the dominant configuration. Reviewing Table 2, it is probably this feature, which causes the deviation in the adiabatic excitation energies between DFT and CASPT2.

Proceeding to **4**, shown in Figure 7, we find a notably different behaviour when compared to isomeric **3**. Viewing the shielding tensors in the ground state, Figure 7a, we find enhanced antiaromaticity at the CBD ring showing deshielding of +61.1 ppm, which is even higher than for isolated CBD (Figure 4b). This enhanced antiaromaticity agrees well with the higher relative energy as shown in Table 2. Both benzene rings come out as clearly shielded, albeit with lower values than isolated benzene. Proceeding to the triplet state (Figure 7b), we find a strongly aromatic CBD ring (−28.6 ppm) similarly to biphenylene. This is accompanied by a weakly shielded intermediate benzene ring and a strongly shielded (−25.1 ppm) outer benzene ring. Reviewing the shielding tensors, these can be seen to reflect the sextet-quartet-sextet structure indicated in Figure 2. It is precisely this feature that has been associated with enhanced stability for triplet excited states [45].

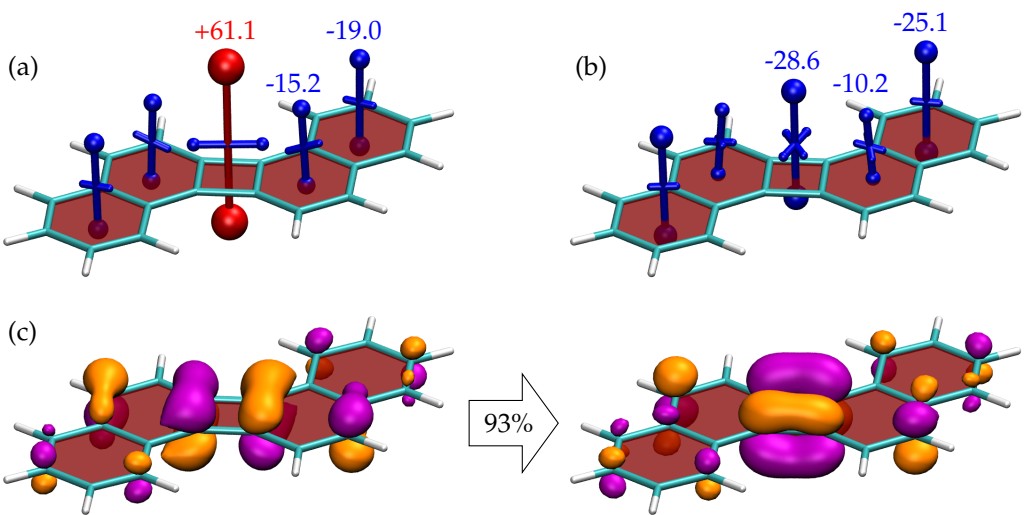

**Figure 7.** Visualisation of chemical shielding tensors (VIST) for (**a**) the $S_0$ and (**b**) the $T_1$ states of **4** computed 1 Å above the molecular plane. The dominant pair of natural transition orbitals (isovalue 0.05 a.u., weight 93 %) for the $S_0 \rightarrow T_1$ transition is shown in (**c**).

Viewing the NTOs of the $S_0 \rightarrow T_1$ transition, we find that, as opposed to **3**, there is only one dominant NTO pair already making up 93% of the overall transition. Moreover, this NTO pair closely resembles the one in biphenylene. In particular there is a strong contribution between the quasi-degenerate CBD frontier orbitals, which is linked with the Baird aromaticity of this compound, as discussed above.

Proceeding to **5**, shown in Figure 8, we find a largely similar structure in terms of, both, the shielding tensors and NTOs as for **4** and, again, entirely different properties to **3**. This highlights that the qualitative considerations in terms of quartets and sextets, as presented in Figure 2, are a powerful way to estimate the properties of these molecules and that other details in the bonding pattern only play a secondary effect.

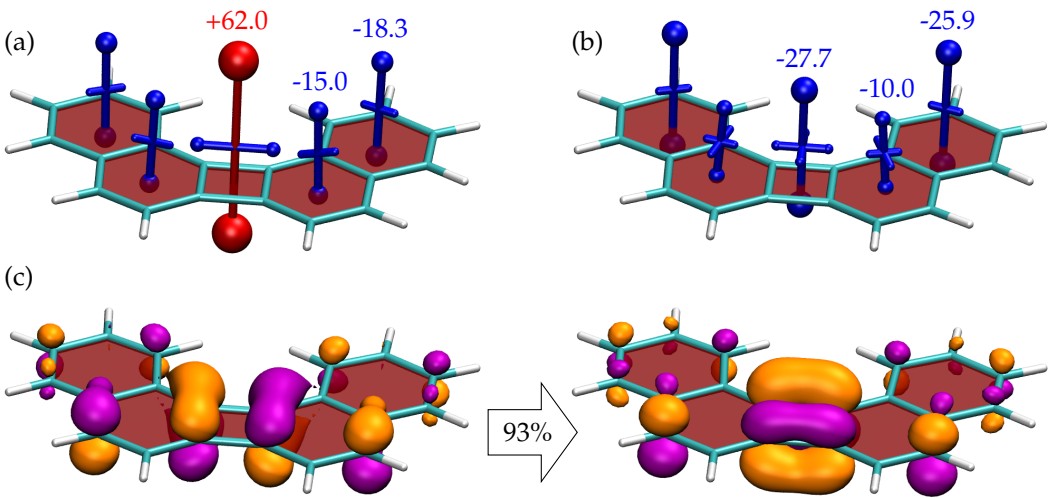

**Figure 8.** Visualisation of chemical shielding tensors (VIST) for (**a**) the $S_0$ and (**b**) the $T_1$ states of **5** computed 1 Å above the molecular plane. The dominant pair of natural transition orbitals (isovalue 0.05 a.u., weight 93 %) for the $S_0 \rightarrow T_1$ transition is shown in (**c**).

## 4. Discussion

Before concluding, we want to discuss the relation between the chemical shielding tensors used above and other common ways of describing the electronic structure of PAHs. In the above presentation, we have shown that Baird aromaticity provides a powerful way of rationalising variations in the triplet energies of the molecules studied, similarly to the discussion in [45] where molecules **2**–**5** have also been analysed. In addition, we have highlighted that the energies can also be explained within the MO picture following Figure 3 and by realising that the orbitals involved do indeed possess the shapes expected from the rigid rotor model. Using a different viewpoint, Ryerson et al. [27] have pointed out that also a biradical model is highly suitable for discussing the energetics in related systems. It should, therefore, be stressed that Baird aromaticity is just one possible language of describing the phenomena seen. Nonetheless, it is the strength of the aromaticity rules that they allow qualitative predictions based on simply counting the number of electrons and analysing bonding patterns as exemplified in the striking differences between molecules **3** and **4**/**5**.

From a more formal viewpoint it is worth pointing out that there is a specific hierarchy between frontier orbitals, biradical character/NTOs, and the shielding tensors. Frontier orbitals are simply intermediates in approximate theories and it is well-known that their shapes and energies change dramatically with the level of theory chosen [55,94]. Measures for biradical character and the NTOs can be defined based on the wavefunctions alone [24,76,95] and are, thus, well-defined without any explicit reference to a computational level of theory. However, out of this list, only the chemical shielding tensors are a physically observable quantity. The chemical shielding at the atomic positions is of course routinely measured in [1]H- and [13]C-NMR experiments. Even more, it is possible to approximate the NICS via NMR experiments using complexated [6]Li [96] and, more generally, the shielding defined as a ratio of two magnetic fields (Equation (1)) is always at least in principle an observable.

More generally, we suggest using aromaticity as one viewpoint in the elucidation of excited-state electronic structure in a similar sense as exciton theory and valence bond theory can be helpful viewpoints in elucidating otherwise hidden properties of excited states [93,97]. However, it should be understood that these are just different ways of describing the same underlying physics and that Baird aromaticity is not somehow detached from other ways of describing excited-state electronic structure.

The present work illustrates the molecules studied from three independent viewpoints: energetics, NTOs, and shielding tensors. All three measures provide consistent trends suggesting that the Baird aromaticity model is, indeed, a powerful way of describing the

underlying physics. In this context it should be pointed out that the practice of computing only one NICS value per ring in multi-ring systems, as done above, has been contested [58] as it does not provide a complete picture of the magnetic properties of the molecule studied meaning that more fine-grained scans [98] may be needed. The shielding tensors certainly do not contain as much information as the full underlying current density vector field [99]. Even more, the ring currents themselves are not always a reliable measure for aromaticity [65]. The VIST method is certainly not able to overcome any of these problems and has to be used with caution just as any other computational method. Conversely, we believe that for the purposes of the current study, i.e., comparing the electronic structure of closely related systems and illustrating the excitation processes, it provides an intuitive and compact way representing electronic structure properties.

## 5. Conclusions

Within this work a class of PAHs based on the biphenylene motif was studied showing that the bond topology had dramatic effects on their triplet excitation energies. Strikingly, a difference of more than 1 eV in adiabatic excitation energies was found for **3** when compared to isomeric **4** and **5**. These results were rationalised in the context of Baird aromaticity using a qualitative description of Clar sextets and Baird quartets, quantified in terms of NICS values, and graphically represented using a recently developed method for visualising chemical shielding tensors (VIST). A comparison with natural transition orbitals showed a consistent picture highlighting that Baird aromaticity should not be seen as a detached phenomenon but as one possible way to categorise excited states. More generally, we believe that excited-state aromaticity constitutes a fascinating and important phenomenon and hope that this work will stimulate future studies by showing how it can be readily visualised and how it integrates into the framework of excited-state electronic structure theory.

**Supplementary Materials:** The following are available online at https://www.mdpi.com/2624-8549/3/2/38/s1; Figures S1–S4: Natural transition orbitals of **2–5** computed with TDDFT ($T_1$ geometry) and CASPT2 ($S_0$ and $T_1$ geometries; Figure S5: Frontier orbitals of naphthalene.

**Funding:** This research was funded by the EPSRC, grant number EP/V048686/1.

**Institutional Review Board Statement:** Not applicable.

**Informed Consent Statement:** Not applicable.

**Data Availability Statement:** Supporting research data available: Molecular coordinates; input/output files for Q-Chem, OpenMolcas, Gaussian, and TheoDORE, doi:10.17028/rd.lboro.14139824.

**Acknowledgments:** The author thanks F. Glöcklhofer for comments on the manuscript.

**Conflicts of Interest:** The author declares no conflict of interest.

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
