# Peer review of "Exploitation of Baird Aromaticity and Clar’s Rule for Tuning the Triplet Energies of Polycyclic Aromatic Hydrocarbons"

_chemistry, doi:10.3390/chemistry3020038_

Round 1

Reviewer 1 Report

In this article, Plasser investigates the aromaticity in ground and excited states of a few polycyclic aromatic hydrocarbons (PAHs) formed from the fusion of cyclobutadiene and benzene rings. The main method used is the recently developed visualising chemical shielding tensors (VIST) that the author has presented and validated in a separate paper available on the ChemRxiv (10.26434/chemrxiv.13580885.v2). VIST gives a clear graphical representation of the chemical shielding tensor and is a more informative (and more complex) alternative to the conventional NICS approach. The analysis is further strengthened by calculation of natural transition orbitals, which reveals the character of the excitations. There is also an interesting conceptual model for the diatropic and paratropic contributions of aromatic and antiaromatic molecules based on the eigenstates of a 2D rigid rotor, which qualitatively explains the different magnetic properties. The paper is overall well written, conceptually clear, and the science is sound. In particular, I liked the pedagogical presentation. As for novelty, the paper is a demonstration of the merit and applicability of the VIST method. The analysis in terms of natural transition orbitals is also a useful addition to the field which mostly has focused on analysing excited state character in terms of spin densities and frontier molecular orbitals. The main conclusions are a corroboration of the conclusions of Ottosson and co-workers on the validity of using a combination of Clar’s and Baird’s rules for describing the triplet states of certain PAHs. I recommend publication after minor revisions:

  1. At what geometry are the TD-DFT calculations done? If ground state were used, the natural transition orbitals obtained would correspond to vertical excitations, while the computed triplet structures and adiabatic energy gaps would correspond to relaxed triplet states. It is not certain that the analysis of the excited state character will hold upon relaxation and it would be better to confirm this with natural transition orbitals also at the relaxed geometries (could be put in the Supporting Information)
  2. It should perhaps be pointed out that the analogy to the rigid rotor eigenstates only applies to the filled molecular orbitals.
  3. The argument on page 6 on the effect of ring size on shielding makes some assumptions and then on the basis of these assumptions conclude that the paramagnetic contribution is insensitive to ring size. These assumptions should either be corroborated or supported by references, otherwise the conclusion is speculative.
  4. I’m wondering if the character of the triplet compound 3 (Figure 6) could be understood in terms of a resonance hybrid, which borrows 28% of its character from (a perturbed) biphenylene and 57% of its character from (a perturbed) naphthalene

Suggestions for future improvements (does NOT need to be implemented in the current manuscript):

  1. Indicate the center of the dumbbell graphically. This would be useful in cases such as benzene where there is only one sizeable eigenvector.

Author Response

I thank the referee for pointing out the merits of the manuscript in terms of providing a conceptual model for understanding diatropic and paratropic contributions to the chemical shielding, evaluating the merits of the VIST method, and comparing NICS to natural transition orbitals.

1. At what geometry are the TD-DFT calculations done? If ground state were used, the natural transition orbitals obtained would correspond to vertical excitations, while the computed triplet structures and adiabatic energy gaps would correspond to relaxed triplet states. It is not certain that the analysis of the excited state character will hold upon relaxation and it would be better to confirm this with natural transition orbitals also at the relaxed geometries (could be put in the Supporting Information)

Response: Yes, the NTOs in the main manuscript correspond to the vertical excitation (S0 geometry). This is now stated explicitly in the Computational Details. NTOs at the T1 geometry have been added to the Supporting Information following the referee's suggestion. These possess very similar shapes as the ones at the S0 geometry. In addition, CASPT2 NTOs have been computed, also possessing a similar shape.

2. It should perhaps be pointed out that the analogy to the rigid rotor eigenstates only applies to the filled molecular orbitals.

Response: Generally speaking, all the orbitals except for the last one should comply with the rigid rotor symmetry. We have added an explanation:

Only the highest, fully anti-bonding pi*-orbitals of benzene and CBD (not shown in Fig.~3) do not fit the analogy, since they do not have a degenerate partner.

3. The argument on page 6 on the effect of ring size on shielding makes some assumptions and then on the basis of these assumptions conclude that the paramagnetic contribution is insensitive to ring size. These assumptions should either be corroborated or supported by references, otherwise the conclusion is speculative.

Response: Thanks to the referee for pointing this out. I have deleted the discussion of the paramagnetic contributions, since it is too speculative and not actually needed for the remainder of the manuscript.

4. I’m wondering if the character of the triplet compound 3 (Figure 6) could be understood in terms of a resonance hybrid, which borrows 28% of its character from (a perturbed) biphenylene and 57% of its character from (a perturbed) naphthalene

Response: This is an interesting question. But I do not think that there is an easy way to map from MOs to resonance structures. Resonance structures appear within valence bond theory, which builds on different ideas than MO theory.

Future improvement: Indicate the center of the dumbbell graphically. This would be useful in cases such as benzene where there is only one sizeable eigenvector.

Response: This is an interesting suggestion, which will be considered for a future release of TheoDORE. At the moment the priority is to release a documented and tested version of the existing functionality.

Reviewer 2 Report

Aromatic properties of the triplet state of polycyclic aromatic hydrocarbons have been studied by calculating and visualizing magnetic shielding tensors. The rigid rotor viewpoint is not new. The author could read about it in Spectrochimica  Acta Part A 55 (1999) 585–606, which is not the first paper. Michl publish it already in the 70ties. The author can search for the original suggestion, discuss it and cite it. The author mentions current density calculations and completely ignore the Italian groups (Lazzeretti, Zanasi). The author cite Fliegl and Geuenich in that context, which seem to be some random studies that the author is aware of. Original work should be cited. The NICS paper is a review from 2005, whereas the original NICS paper is not cited.  Calculating magnetic shieldings using gauge-including atomic orbitals means that the most of the expressions in section 2.2 are incorrect. The author writes ''Practically evaluating this expression leads to the following form'' and cite doi:10.1063/1.471789. The author should read cited references. The expression for evaluating magnetic shieldings are completely different. The author  cites something as it does not matter what one cites.  Pulay's work changed the methods for calculating magnetic shieldings and it is not mentioned. Limitations of NICS calculations have been discussed in many papers.  The present method is an extension of the NICS method involving the whole tensor.  It suffers from same kind of problems as NICS calculations and the author has decided to ignore that fact in the paper. The NICS values of  the individual rings of molecule 2 are smaller than the NICS values of benzene and cyclobutadiene. The author assumes that the NICS values originate from the ring current of the rings. In the triplet state, the NICS values are all negative and large suggesting aromaticity. In the paper one can read ''These changes highlight the Baird aromaticity of the triplet state in line with the NTOs presented.'' What does it mean? Is Baird a global property or does each ring have a ring current? From symmetry considerations one can conclude that the individual rings cannot have ring currents of their own.  The employed computational method has the same limitations as NICS calculations and cannot be used for understanding the studied molecules. In the NICS society one has introduced 2D and 3D shielding functions to understand the aromaticity of complicated molecules.  Triplet states have been studied at the DFT level, which is problematic. Are similar results  obtained at ab initio correlated levels of theory.

Author Response

Aromatic properties of the triplet state of polycyclic aromatic hydrocarbons have been studied by calculating and visualizing magnetic shielding tensors. The rigid rotor viewpoint is not new. The author could read about it in Spectrochimica  Acta Part A 55 (1999) 585–606, which is not the first paper. Michl publish it already in the 70ties. The author can search for the original suggestion, discuss it and cite it.

Response: Following the reviewer's suggestion, we have added these references at the beginning of Section 2.1.

The author mentions current density calculations and completely ignore the Italian groups (Lazzeretti, Zanasi). The author cite Fliegl and Geuenich in that context, which seem to be some random studies that the author is aware of. Original work should be cited. The NICS paper is a review from 2005, whereas the original NICS paper is not cited.

Response: A Lazzeretti et al. reference has been added and the original NICS paper is now cited.

The present manuscript cites review/perspective articles describing the GIMIC, ACID etc methods, thus, properly crediting the authors and giving interested readers a place for further reading.

Calculating magnetic shieldings using gauge-including atomic orbitals means that the most of the expressions in section 2.2 are incorrect. The author writes ''Practically evaluating this expression leads to the following form'' and cite doi:10.1063/1.471789. The author should read cited references. The expression for evaluating magnetic shieldings are completely different. The author  cites something as it does not matter what one cites.  Pulay's work changed the methods for calculating magnetic shieldings and it is not mentioned.

Response: I thank the referee for pointing out this possible misunderstanding. The relevant part in Section 2.2 has been extended to allow to clearly differentiate between the numerical calculations performed using gauge-including AOs and the qualitative model used to explain them.

This distinction, however, is not unique and the relative magnitude of these terms depends on the gauge origin chosen [58]. For practical calculations, it is expedient to include the gauge definition directly within the atomic orbitals [56, 61]. However, for the following qualitative discussion, we will use a different convention setting the gauge origin to the point where the shielding tensor is computed ($\vec{R}=\vec{0}$). Under this approximation, the diamagnetic term -- exemplified by one of its diagonal components $\sigma_{zz}^\mathrm{dia}$ -- can be rewritten as

Pulay's reference is now included.

Limitations of NICS calculations have been discussed in many papers.  The present method is an extension of the NICS method involving the whole tensor.  It suffers from same kind of problems as NICS calculations and the author has decided to ignore that fact in the paper.

Response: A discussion of the limitations was already present in the last  paragraph of the Discussion section. We have extended this paragraph in the revision.

The NICS values of  the individual rings of molecule 2 are smaller than the NICS values of benzene and cyclobutadiene. The author assumes that the NICS values originate from the ring current of the rings. In the triplet state, the NICS values are all negative and large suggesting aromaticity. In the paper one can read ''These changes highlight the Baird aromaticity of the triplet state in line with the NTOs presented.'' What does it mean? Is Baird a global property or does each ring have a ring current? From symmetry considerations one can conclude that the individual rings cannot have ring currents of their own. 

Response: The reviewer highlights a problem regarding the interpretation of ring currents: The net ring current between symmetry-equivalent rings vanishes. This is not because the electrons are somehow completely stationary but because currents in both directions cancel out. Therefore, we have not used ring currents in our explanations. Section 2.2 directly explains the NICS values in terms of symmetry properties of the frontier orbitals, which avoids the problems the reviewer mentions.

The employed computational method has the same limitations as NICS calculations and cannot be used for understanding the studied molecules. In the NICS society one has introduced 2D and 3D shielding functions to understand the aromaticity of complicated molecules.

Response: The reviewer is correct that more complicated molecules might require a more complicated analysis. Within the present work, we find good consistency between adiabatic excitation energies, NICS values, and natural transition orbitals. Considering the fact that three distinct ways of viewing the electronic structure point into the same direction, we believe that a consistent picture is achieved. It is not clear what 2D or 3D shielding functions are required for, here.

Triplet states have been studied at the DFT level, which is problematic. Are similar results  obtained at ab initio correlated levels of theory.

Response: Following the reviewer's suggestion, we have included ab initio CASPT2 computations on the singlet and triplet states of all molecules involved. Good agreement is obtained in terms of energies (see Table 1) and orbitals (see Figures S1-S4) suggesting that the DFT description is indeed sufficient.

Reviewer 3 Report

Author of this work analyzes triplet state energies of the three isomeric dibenzobiphenylenes and explains them in terms of (1) the Huckel/Baird aromaticity rules combined with the Clar’s rule, (2) the recently developed VIST method (by the same author and co-worker) based on magnetic shielding tensor and (3) shape of HOMO/LUMO orbitals. The first one has already been investigated and explained in ref. 41, so that, in my opinion, this should not be stressed as the focus of this work, just the last two, which are nicely presented and discussed. Thus, I recommend the manuscript for publication after minor revision.

  1. It should be clear, throughout the text, that connection between the triplet state energies and Clar’s and Baird’s rules for the studied molecules has also been analyzed in detail in ref. 41.
  2. In my opinion, it would be useful to include naphthalene orbitals on p. 10, where they are mentioned.

Minor:

  1. p. 3, line 78: “are shown” is doubled.
  2. p. 10, line 305: “middle” instead of “bottom”?
  3. Ref. 73: the year is 2020.

Author Response

I thank the referee for his/her positive assessment.

It should be clear, throughout the text, that connection between the triplet state energies and Clar’s and Baird’s rules for the studied molecules has also been analyzed in detail in ref. 41.

Response: The reviewer is correct that Ref. 45 (41 in the initial submission)   discusses molecules 2-5 in the context of Baird aromaticity. We have mentioned this more explicitly in the Introduction, Results, and Discussion sections.

In my opinion, it would be useful to include naphthalene orbitals on p. 10, where they are mentioned.

Response: Following the reviewer's suggestion, these orbitals have been added to the Supplementary Information (Fig. S5).

3. p. 3, line 78: “are shown” is doubled.

Response: Corrected.

p. 10, line 305: “middle” instead of “bottom”?

Response: No, it should be the orbital without nodal planes that is shown at the bottom.

Ref. 73: the year is 2020.

Response: Corrected.

Round 2

Reviewer 2 Report

The author has considered suggestions of the referees. However, I suggest the following changes before it can be accepted for publication.  The new discussion on lines 163-177 may not be completely correct.  See reference 50, where they show that the number of electrons contributing to the current  density is 2 and 4 for aromatic and antiaromatic rings, respectively. A large number of calculations shows that the ring current strength is rather independent of the number of pi electrons and the size of the ring.  At least no linear relation has been found. It has been studied in more detail. See https://pubs.acs.org/doi/10.1021/jp411194m and references  therein. Reference 64 is a very limited study of six N6H6(2+) structures. One has 8 imaginary frequencies, two high-lying ring structures and 3 open structures. I would not support a conclusion based on such a  limited study. Umlaut letters should be added to names in the references.

Author Response

The author has considered suggestions of the referees. However, I suggest the following changes before it can be accepted for publication.  The new discussion on lines 163-177 may not be completely correct.  See reference 50, where they show that the number of electrons contributing to the current  density is 2 and 4 for aromatic and antiaromatic rings, respectively. A large number of calculations shows that the ring current strength is rather independent of the number of pi electrons and the size of the ring.  At least no linear relation has been found. It has been studied in more detail. See https://pubs.acs.org/doi/10.1021/jp411194m and references  therein.

Response: The reference suggested by the referee has been added (new Ref. 50).

The presented discussion in lines 163-177 agrees well with two phenomena seen in Ref. 57: (i) NICS values are insensitive to the ring size and (ii) aromatic dianions have enhanced NICS values when compared to aromatic dications. Therefore, I believe it is instructive to keep this discussion.

Interestingly, the very reference suggested by the referee seems to indicate a linear relation between pi-signed current strengths and the number of electrons (see Table 1 of Ref. 50). Out of the CnHn systems one finds about -4nA/T for 2-el systems, -12nA/T for 6-el systems, and -20nA/T for 10-el systems.

Reference 64 is a very limited study of six N6H6(2+) structures. One has 8 imaginary frequencies, two high-lying ring structures and 3 open structures. I would not support a conclusion based on such a  limited study.

Response: I have made an effort to discuss the limitations of the NICS method following the referee's suggestion and have extended the last paragraph of the Discussion.  Neither Ref. 64 nor the other references mentioned in that paragraph are expected to have an immediate effect on the conclusions made within the paper and this is stated explicitly. Nonetheless, it may be helpful for future users to learn from different viewpoints about possible limitations of the method.

Umlaut letters should be added to names in the references.

Response: I have corrected the Umlaut in Ref. 67. Otherwise, I have checked that Umlaute and accents are displayed appropriately.